# Impacts of COVID-19 on the Food Supply Chain: A Case Study on Saudi Arabia

**Alhanouf Abdulrahman Alsuwailem** [1,*], **Emad Salem** [2], **Abdul Khader Jilani Saudagar** [1,*], **Abdullah AlTameem** [1], **Mohammed AlKhathami** [1], **Muhammad Badruddin Khan** [1] **and Mozaherul Hoque Abul Hasanat** [1]

1  Information Systems Department, College of Computer and Information Sciences, Imam Mohammad Ibn Saud Islamic University (IMSIU), Riyadh 11432, Saudi Arabia; altameem@imamu.edu.sa (A.A.); maalkhathami@imamu.edu.sa (M.A.); mbkhan@imamu.edu.sa (M.B.K.); mhhasanat@imamu.edu.sa (M.H.A.H.)
2  Department of Statistics, Institute of Public Administration, Riyadh 11141, Saudi Arabia; emadsalem@gmail.com
*  Correspondence: aasalsuwailem@imamu.edu.sa (A.A.A.); aksaudagar@imamu.edu.sa (A.K.J.S.)

**Abstract:** The entire world is suffering from the post-COVID-19 crisis, and governments are facing problems concerning the provision of satisfactory food and services to their citizens through food supply chain systems. During pandemics, it is difficult to handle the demands of consumers, to overcome food production problems due to lockdowns, work with minimum manpower, follow import and export trade policies, and avoid transportation disruptions. This study aims to analyze the behavior of food imports in Saudi Arabia and how this pandemic and its resulting precautionary measures have affected the food supply chain. We performed a statistical analysis and extracted descriptive measures prior to applying hybrid statistical hypothesis tests to study the behavior of the food chain. The paired samples *t*-test was used to study differences while the independent samples *t*-test was used to study differences in means at the level of each item and country, followed by the comparison of means test in order to determine the difference and whether it is increasing or decreasing. According to the results, Saudi Arabia experienced significant effects on the number of items shipped and the countries that supplied these items. The paired samples *t*-test showed a change in the behavior of importing activities by—47% for items and countries. The independent *t*-test revealed that 24 item groups and 86 countries reflected significant differences in the mean between the two periods. However, the impact on 41 other countries was almost negligible. In addition, the comparison of means test found that 68% of item groups were significantly reduced and 24% were increased, while only 4% of the items remained the same. From a country perspective, 65% of countries showed a noticeable decrease and 16% a significant increase, while 19% remained the same.

**Keywords:** food supply chain; COVID-19; pandemic; imports; hypothesis tests

## 1. Introduction

Food security in Saudi Arabia is aligned with the global market in order to manage its food security. The economy of Saudi Arabia is strong enough to allow it to import the required food by establishing market-based food security. To meet the needs of the population, it imports approximately 80% of its food requirements [1]. According to a sustainability development program launched by the Saudi Government, Saudi Arabia is working on creating an excellent sustainable and responsible supply chain by conducting advanced planning systems to optimize goods flow, full inventory visibility through the technology, development of infrastructure to improve the transportation efficiency, logistics modeling that employs tools to enhance warehouse and terminal; location, size and transportation among distributors, and communication and training programs to build employees skills in this field [2]. To clearly define the meaning of food security at the national level, the Food and Agriculture Organization (FAO) stated the following: "all

people, at all times, have physical and economic access to sufficient, safe and nutritious food to meet their dietary needs and food preferences for an active and healthy life" [3]. The following three main measures are relied upon to obtain a better understanding of food security: food availability, which measures the sufficiency of quantities; food access, which considers the ability of households to obtain the required quantities; and food utilization, which focuses on the cleanliness and nutrition of suitable food [4].

By contrast, the risk is raised by a market-based food supply that food security may be affected due to urgent events that happen suddenly around the globe, such as reductions in the supply of food exports due to natural disasters; thus, countries that are mainly dependent on imports will be most at risk of food insecurity [1]. The coronavirus pandemic (COVID-19) is considered one such food-security risks, as it also affects economies and activities dramatically [4–6], including manufacturing and supply chain logistics. Moreover, it influences the food supply chain around the world, as well as food production [4,5] and transportation capacity [7]. In the study conducted by [8], the top six Asian countries affected by COVID-19 on their exports and logistic performance include Pakistan, Bangladesh, Iran, India, Indonesia, and Philippines. The study found strong negative association of COVID-19 on exports and logistic performance. It is worth mentioning that the focus in this research is on the food supply chain specifically, not on transportation issues in the food supply chain in Saudi Arabia. Given that Saudi Arabia has limitations in terms of agricultural production, it frequently imports food from other countries [4]. The global food security index in 2020 showed that Saudi Arabia ranked 38th and obtained a score of 69.5/100, while in 2019, the country ranked 30th and garnered a score of 73.5/100 [9]. The burden of COVID-19 on food supply chains and the resilience of global food imports during the pandemic in countries such as Saudi Arabia, must be explored. This paper exhibits how import activities are performed while applying precautionary measures in order to obtain insights and understand the impacts of the pandemic on food supply chains.

The motivations for this study are highlighted as follows:

1. To discuss the behavior of food imports in Saudi Arabia during COVID-19 and how this pandemic and its resulting precautionary measures affected the country's food supply chain.
2. Due to the lack of availability of food supply chain data, particularly about Saudi Arabia, this primary data was received from SFDA and not used elsewhere to conduct this study.
3. To obtain a better understanding, we performed descriptive analysis to identify the most importing food items and from where the items were imported.
4. In this regard, a statistical test on real imported food items has been performed to explain the impact of COVID-19 on the food supply chain.

This paper aims to discuss the behavior of food imports in Saudi Arabia during COVID-19 and how this pandemic and its resulting precautionary measures have affected the food supply chain of the country. We performed a statistical test on real imported food items in Saudi Arabia. The rest of the paper is constructed as follows: the literature review section discusses previous studies conducted on food supply chains. The list of previous studies was obtained from the Saudi Digital library (SDL) by searching recent publication years using specific keywords: supply chain, food supply chain, food sustainability, and impact of COVID-19. Section 2 then explains the methodology used to conduct this study. Sections 3 and 4 present the data collection and preparation steps, Section 5 provides the results of the statistical analysis, and Section 6 concludes this paper.

## 1.1. Food Supply Chain

The network of organizations that are connected upstream (i.e., supply) and downstream (i.e., distribution) by applying different processes and activities to deliver a product or service to the final consumer is known as a supply chain [10]. A food supply chain is defined as "a series of links and inter-dependencies from farms to food consumer plates, embracing a wide range of disciplines" [11]. Thus, food proceeds from food creators to con-

sumers by the following five stages: production processes, processing, allocation, retailing, and consumption [12]. There are three different types of supply chains: direct, extended, and final. The simplest supply chain is the direct supply chain, because it has a supplier, a central company, and a buyer, while an extended one includes a supplier, a company, a buyer, and the supplier of a supplier. Further, the final supply chain covers all goods, services, information, and capital flows from the first supplier until the final consumer [13]. M. Bourlakis and Weightman [14] mentioned six main factors which have an effective role in food supply chains development: (1) Quality is considered as a degree of harmony among expectations of consumers and their fulfillment; (2) Technological development consists of the innovations and developments which allow for integrity, efficiency, and productivity, e.g., accurate weighing, refrigeration, barcoding; (3) Logistics is knows as 'the process of strategically managing the procurement, movement, and storage of materials, parts and finished inventory via an organization and its marketing channels in such a way that current and future profitability are maximized through cost-effective fulfillment of orders'; (4) Information technology helps a product's movement and the dissemination of related information, e.g., using bar codes to identify the products by using optical methods; (5) The regulatory framework consists of national and international laws concerning the safety, labeling, and traceability of food products; and (6) Consumers, who drive the supply chain because the manufacturers, wholesalers, and retailers all work to achieve the demands of consumers in a better and more efficient manner.

Consumers are paying more attention to food supply chains as they are worried about outbreaks that arise from disease and shortage issues [15]. The outbreak of COVID-19 has brought a global disaster in terms of human lives, manufacturing, and supply chains. Food supply chains for products such as grains, fresh vegetables, fruits, and bakery products have been sharply influenced by the pandemic [16].

### 1.2. Influence of COVID-19 on Food Supply Chains

COVID-19 has directly influenced food supply chains and agriculture in two major ways: food supply and food demand. These two perspectives directly affect food security; during the start of the COVID-19 pandemic, panic buying left empty shelves at supermarkets and stores because of sudden changes in demand [5]. One affected sector is the food industry. The consumption of food and goods has increased, as people have tended to stock them up during quarantines and lockdowns [17]. The demand for food has increased around the globe, as demonstrated by information provided by European nations; specifically, demand increased within the week that COVID-19 was first announced, by 76% for bread and 52% for vegetables [5]. Additionally, flour has been considered the product most highly demanded in the UK; every 14 weeks, households bought 1.5 kg of flour during lockdown periods [17]. In France, the use of trucks to distribute food decreased to 60% due to restrictions, which was 30% before COVID-19 [12]. Further, in food-processing plants, production has declined as workers have been infected with COVID-19 or have been resistant to going to work due to the virus [12].

The restrictions made by governments have affected sea, air, and land transportation and the distribution of food to suppliers, retailers, and consumers. Moreover, curfews have affected items and financial flows as well [6]. The consumption of food in Germany during the pandemic has influenced the volume of transport and the capacity levels of freight for food retail logistics. After performing regression analysis, we found that when the transport volume increases, the number of COVID-19 cases also grows. Thus, in Germany, the demand for food is mainly based on the volume of outbreak (number of cases) and not on the duration of the pandemic [17]. Seasonal employment is highly common in developing countries for the planting, harvesting, sorting, processing, or transporting of crops. Therefore, the absence of this type of workers due to lockdowns or sickness has significantly affected supply chains [12].

In this context, in order to ensure food security, the continuity of food production and the availability of ingredients are essential. Food production has been affected by

the lockdowns and precautionary measures applied by various governments. In China, it has been found that strict precautionary measures imposed by the government have numerous impacts in terms of delaying production, restricting agricultural products, and damaging the cycles of production [17]. In addition, COVID-19 restrictions have affected the number of container ships (8%, which is below normal). This impact has resulted from the limitations imposed in terms of staff changes, more screening, quarantines, and reduced demand. In Canada and the United States, road transport decreased by 20% in April 2020, which is below normal [18].

### 1.3. Challenges of Food during the COVID-19 Pandemic

Food supply chains and food industries are at risk of being disrupted by global crises [17] as COVID-19 has consequences on the agribusiness sector. In Brazil, the price of food differs between regions affected by COVID-19. A food distribution center has found a correlation between food prices and the gravity of the situation in regions affected by COVID-19. The price of tomatoes has increased by 66.91%, and that of onions by 101.53% [17].

The pandemic has effects on both demand and supply. On the demand side, at restaurants and other food companies, large decreases in food consumption have emerged, while sales in grocery stores have increased suddenly due to panic buying, thereby increasing the pressure on the food system. Further, the supply side has been mainly affected by the shortage of laborers caused by absenteeism from work, illnesses, and fear of COVID-19, thereby affecting the speed of operations, especially in concentrated industries such as the meat packing industry [19].

### 1.4. Strategies for Food Supply Chains

The tool called "Supply Chain Management (SCM) Data Science" is employed to consider SCM problems and predict results by applying qualitative and quantitative techniques that consider the quality and accessibility of information. Utilizing the right data to improve supply chains is crucial to working efficiently. Consequently, accessing reliable information will assist in minimizing uncertainties and identifying potential risks and disruptions. Hence, the right data help in making good decisions, thereby improving profitability [5,12]. Knowledge management (KM) is mainly used to support making decisions by sharing, compiling, and supervising all the practices related to the information in order to meet the goals and techniques of businesses and to reach capacities and abilities [5].

However, the pandemic has influenced the food industry and supply chains by issuing five main challenges: food safety, production, logistics, pricing, and food system survivability. The suggested methods for food safety are developing and utilizing bioanalytical tools to detect the presence of COVID-19 in food, people, and surrounding environments. Conversely, the control of the restrictions and measures applied by governments should not contradict the transportation of primary products and workers. In addition, developing new strategies for manufacturing and looking for alternative ingredients that can be used temporarily when the acquisition of real ingredients is difficult. Further, in a logistical challenge, the coordination between government and private organizations is necessary to develop policies that allow essential goods to move freely [17].

### 1.5. Sustainability in the Food Supply Chain

Sustainability has various definitions, and includes concerns about economics, the protection of the environment, and the resources of ecosystems [20]. The United Nations does not consider the pandemic of COVID-19 to be only a health pandemic, as it has affected food security and the distributed food system around the globe. In addition, the food supply chain has been affected by the pandemic in a negative way. There is concern about the way that the global pandemic in 2020 reveals the challenging of preserving the sustainability of food supplies globally in order to provide food to the increasing population and maintain the environment. There are coordinated efforts to research the rate of food

loss and waste (FLW) that needs to be reduced in order to avoid any decrease in food sustainability [21]. Based on a recent UNCTAD (United Nation Conference Trade and Development) paper, one of the major consequences of the global pandemic is the slowing down of the economy; thus, progress on achieving the 2030 Sustainable Development Goals (SDGs) is greatly affected. Further, there is an opportunity for developing countries to utilize and maximize sustainability in order to improve their trade, which helps to integrate them into the global economy. Therefore, it is important to consider strategic production as the major component which leads to sustainable trade [22]. On the other hand, in a corporate sustainability framework that is more flexible than previous models based on specific criteria, the authors of [20] developed about 45 sustainability sub-criteria based on nine categories: Corporate Governance, Transparency and Communication, Responsibility of Product, Environment, Social, Economics, Natural Environment and Climate Change, Energy Consumption, and Saving; the last new category, Pandemic, is a new criterion due to its importance in developing the model of corporate sustainability.

## 2. Methodology

This case study aimed to analyze the importation of food during the COVID-19 pandemic in order to understand how the food supply chain in Saudi Arabia has been affected by this pandemic. All statistical analyses were carried out by using SPSS software version 25. To analyze the Saudi food data, we used descriptive analysis and independent samples *t*-test as applied by [23]; an additional two tests, paired samples *t*-test and comparison means test, were used to improve the result of analysis. The approach used to achieve the research objectives is shown in Figure 1.

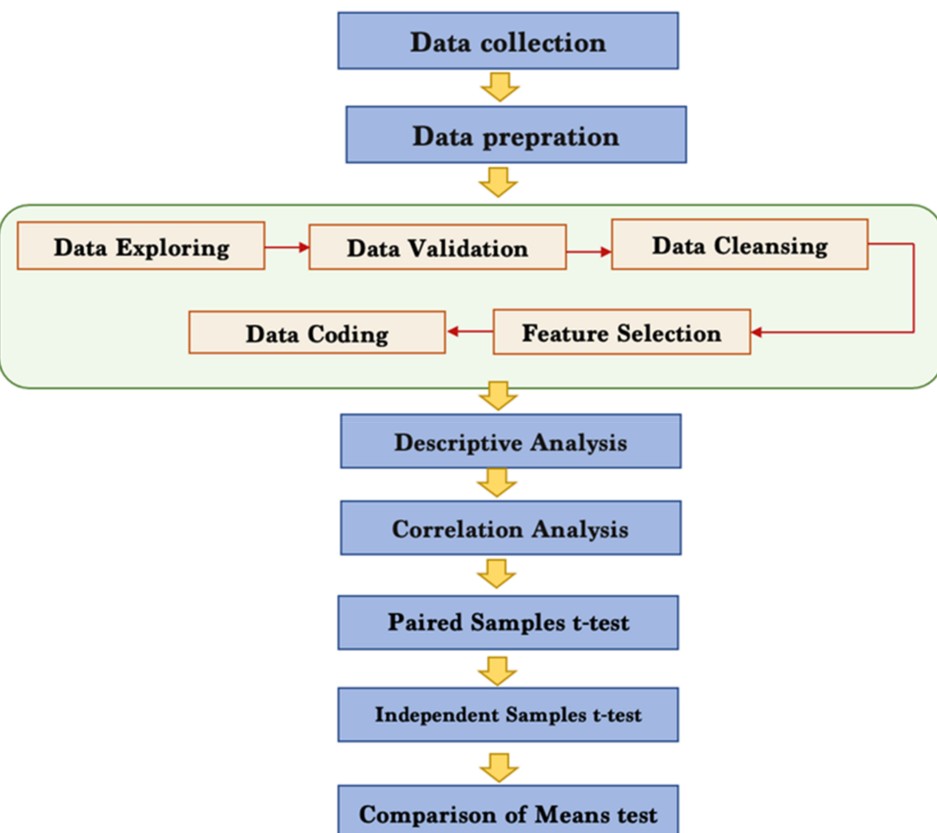

**Figure 1.** The workflow of the case study.

There were two sample cases to observe the behavior of the food supply chain of Saudi Arabia: before the emergence of COVID-19 in Saudi Arabia, and after. All tests were conducted using two dependent variables (test variables); the first variable was the

number of shipments imported for each food item group, and the second was the number of shipments imported from each country. Each variable was tested in two cases, before and after the start of the COVID-19 pandemic.

After performing descriptive analysis, investigating the data, and obtaining more insight into how COVID-19 affects food supply chains, we applied three statistical tests of significance, as shown in Figure 1, in order to make quantitative decisions about the hypothesis. The paired samples *t*-test was applied to measure the difference in the mean in general before and after the emergence of COVID-19. Then, the independent samples *t*-test (also known as the unpaired *t*-test) was applied to compare the differences in the means of the two populations [24]. This is considered a more detailed test to check the difference of mean for each item group and country. The third test was the comparison of means test, which was applied to check in detail whether the difference in mean increased or decreased for each item group and country.

For a valid paired samples *t*-test and independent samples *t*-test, there are two main conditions. First, the distribution should be normal. There should be no extreme outliers [25,26]. To achieve those conditions, we applied the central limit theorem, which assumes that the distribution of a sample means is normally distributed when the sample becomes large [27]. In addition, the independent samples *t*-test requires one more additional condition, homogeneity of variances, which assumes that the variances are equal for the two defined populations [26].

## 3. Data Collection

The data were obtained from the Saudi Food and Drug Authority (SFDA), which regulates, oversees and controls food, drug and medical devices [28]. The original data spanned the period from 1 January 2018 to 9 December 2020, and consisted of 1,048,575 records and 24 features. The total number of items was 25; the most frequent item group was preparations of cereals, flour, starch, or milk, followed by pastry products at 21% and preparation of vegetables, fruits, or nuts at 18%. The total number of countries in the original dataset was 141, and the most frequent was the United States (9%), then the United Arab Emirates and India (6%).

## 4. Data Preparation

### 4.1. Status Feature

In the SFDA dataset, there is a status feature that presents the situation for each shipment item request, which has four types: (1) registered, with 94%; (2) rejected, with 1.70%; (3) inactive, with 0.02%; or (4) deleted by customer, with 4%. We considered the data with 'registered' or 'inactive' status and filtered out those with a status of 'rejected' or 'deleted by customer'.

### 4.2. Harmonized System Code

Harmonized System (HS) codes are standard numerical codes used to classify products, and are used globally by customs authorities. The World Customs Organization (WCO) manages HS codes and updates them every five years [29]. An HS code identification contains a six-digit code with 5000 commodity groups. These groups have 99 chapters, and chapters have 21 sections [30]. In the SFDA dataset, there are 1592 unique HS codes with eight digits. For the clarification of the meaning of codes in an eight-digit structure of HS codes, the first two digits explain the chapter of products, while four digits define the headings. The six digits show the subheadings, and the eight digits define the tariff items [31], as shown in Table 1. The HS code definition is shown on the basis of the digit number identified in a search of tariffs [32].

In the preparation stage, we defined each item in the dataset by using only the first two digits of the HS codes that define the items' chapters. Thus, the total chapters are 55; 30 chapters do not belong to food and were thus removed from the dataset, with 25 retained as shown in Table 2.

**Table 1.** Example of HS code structure.

| n-Digits | HS Code no. | Definition |
|---|---|---|
| Two digits | 19 | Preparations of cereals, flour, starch, or milk; pastrycooks' products |
| Four digits | 1905 | Bread, pastry, cakes, biscuits, and other bakers' wares, whether or not containing cocoa; communion wafers, empty cachets of a kind suitable for pharmaceutical use, sealing wafers, rice papers, and similar products. |
| Six digits | | Sweet biscuits |
| Eight digits | 19053100 | Sweet biscuits; waffles and wafers; sweet biscuits |

**Table 2.** Item groups in the SFDA dataset after preparation.

| Number | HS Code | Item Group | Count | % |
|---|---|---|---|---|
| 19 | 19 | Preparations of cereals, flour, starch or milk; pastry cooks' products | 165,863 | 20.7 |
| 20 | 20 | Preparations of vegetables, fruit, nuts or other parts of plants | 142,661 | 17.8 |
| 21 | 21 | Miscellaneous edible preparations | 100,688 | 12.6 |
| 18 | 18 | Cocoa and cocoa preparations | 81,367 | 10.2 |
| 4 | 4 | Dairy produce; birds' eggs; natural honey; edible products of animal origin, not elsewhere specified or included | 48,228 | 6 |
| 2 | 2 | Meat and edible meat offal | 38,504 | 4.8 |
| 17 | 17 | Sugars and sugar confectionery | 33,443 | 4.2 |
| 3 | 3 | Fish and crustaceans, mollusks and other aquatic invertebrates | 30,291 | 3.8 |
| 9 | 9 | Coffee, Tea, Mate and Spices | 29,280 | 3.7 |
| 11 | 11 | Products of the milling industry; malt; starches; inulin; wheat gluten | 28,971 | 3.6 |
| 10 | 10 | Cereals | 26,461 | 3.3 |
| 15 | 15 | Animal or vegetable fats and oil and their cleavage products; prepared edible fats; animal or vegetable waxes | 18,558 | 2.3 |
| 16 | 16 | Preparation of meat, of fish or of crustaceans, mollusks or other aquatic invertebrates | 14,243 | 1.8 |
| 22 | 22 | Beverages, spirits and vinegar | 13,047 | 1.6 |
| 7 | 7 | Edible vegetables and certain roots and tubers | 11,032 | 1.4 |
| 12 | 12 | Oil seeds and oleaginous fruits; miscellaneous grains, seeds and fruit; industrial or medicinal plants; straw and fodder | 7032 | 0.9 |
| 8 | 8 | Edible fruit and nuts; peel of citrus fruit or melons | 6515 | 0.8 |
| 13 | 13 | Lac; gums, resins and other vegetable saps and extracts | 1535 | 0.2 |
| 24 | 25 | Salt; sulphur, earths and stones; plastering materials, lime and cement | 1436 | 0.2 |
| 23 | 23 | Residues and waste from the food industries; prepared animal fodder | 900 | 0.1 |
| 25 | 35 | Albuminoidal substances; modified starches; glues; enzymes | 531 | 0.1 |
| 6 | 6 | Live trees and other plants; bulbs, roots and the like; cut flowers and ornamental foliage | 212 | 0 |
| 14 | 14 | Vegetable plaiting materials; vegetable products not elsewhere specified or included | 140 | 0 |
| 1 | 1 | Live Animals | 21 | 0 |
| 5 | 5 | Products of animal origin, not elsewhere specified or included. | 11 | 0 |
| | | Total | 800,970 | 100% |

### 4.3. Period of Study

The original dataset spanned from 1 January 2018, until 9 December 2020. To prepare the data for analysis as the researchers investigated the impacts of COVID-19 on food supply chains, we divided the data into two periods, namely, before and after the emergence of the pandemic in Saudi Arabia. This means the period that the country had reported COVID-19 cases and imposed some precautionary measures (Table 3); hence, we can study the change in the variables of the study before and after the emergence of the pandemic.

**Table 3.** Periods' definition.

| Periods | Date | | Duration | Sample Size | |
|---|---|---|---|---|---|
| | **From** | **To** | | **Count** | **Percent** |
| Before the pandemic | 1 January 2018 | 1 February 2020 | 758 days | 650,335 | 81% |
| After the pandemic | 2 February 2020 | 9 December 2020 | 248 days | 150,635 | 19% |
| | Total records | | | 800,970 | 100 |

Finally, after data preparation and transposing the item groups and countries into features, the clean dataset had a final size of 1006 records and 156 features (see Table 4), covering 25 item groups and 127 unique countries.

**Table 4.** Final data features.

| Feature | Description |
|---|---|
| Clearance Close Date | The date where the item clearances and registered to enter Saudi Arabia. |
| Period | At what period the item is imported, before or after the emergence of the pandemic. Before period = 1, After period = 2. |
| Item_group_01 to Item_group_35 | The description of the item group, it represented by 27 features, each item group represented as a specific feature. |
| Country_1 to Country_127 | The country where the shipments are imported from, this column represented by 127 features, each country represents as a specific feature. |

## 5. Descriptive Analysis

After preparing the dataset, the results of the descriptive analysis are demonstrated in this section. The analysis was performed in two stages: (1) the top ten item groups imported (Figure 2) and the top ten countries from which the most item groups were imported (Figure 3).

### 5.1. Correlation Analysis

After checking the correlation of the number of shipments imported for each food item group in both periods, the results show that there is a significantly very strong positive correlation between periods (before and after), with 0.99 and $p$-value < 0.01 (Table 5). This means that the general distribution of the number of shipments of item groups that were imported before the emergence of COVID-19 is positively proportional to the number of shipments of the same item groups that were imported after the emergence of COVID-19, which confirms that the general distribution of the types of item groups was not affected by the emergence of the pandemic.

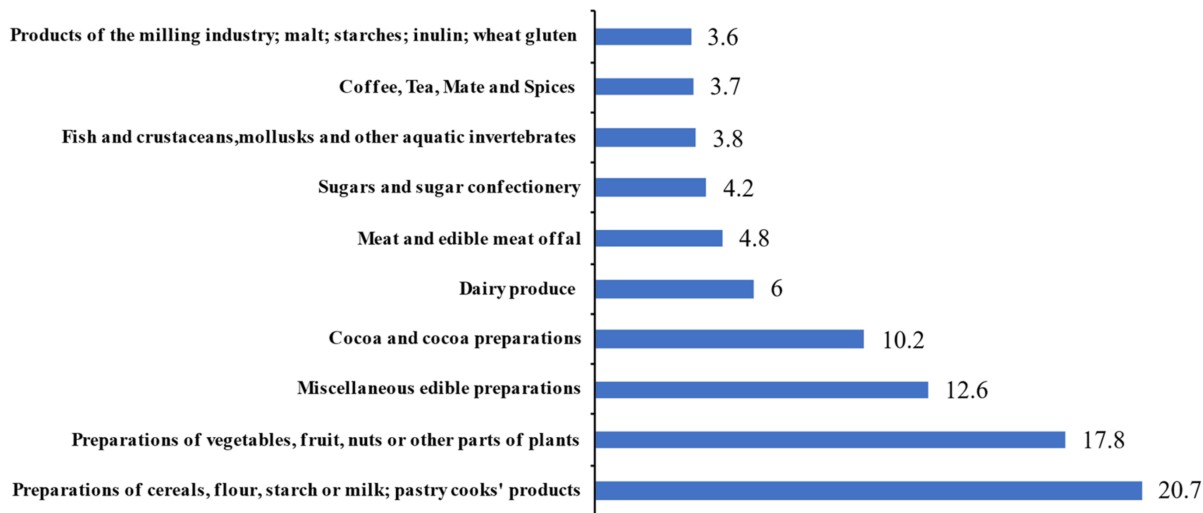

**Figure 2.** Top ten item groups.

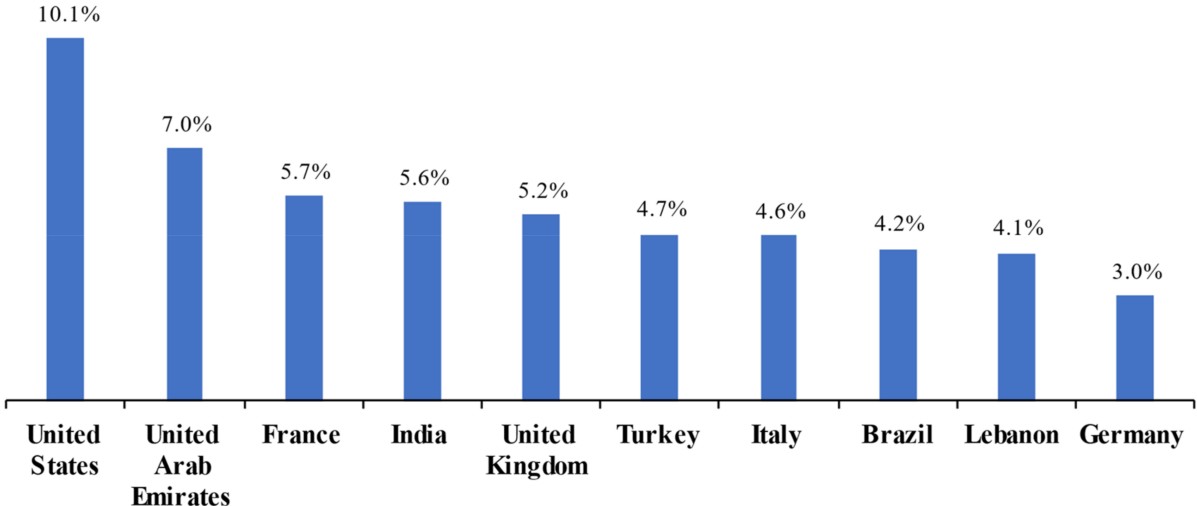

**Figure 3.** Top ten countries.

**Table 5.** Correlation Analysis for item groups.

| | | N | Correlation | Sig. |
|---|---|---|---|---|
| Pair 1 | Item_Before and Item_After | 24 | 0.990 | 0.000 |

We also checked the correlation of the number of shipments imported from each country in both periods; there is a significantly very strong positive correlation between periods (before and after), with 0.98 and *p*-value < 0.01 (Table 6). This means that the general distribution of the number of shipments imported from the countries before the emergence of COVID-19 is positively proportional to the number of shipments imported from the same countries after the emergence of COVID-19, which confirms that the general distribution of the countries from which imports are made has not been affected by the pandemic.

**Table 6.** Correlation Analysis for countries.

|  |  | N | Correlation | Sig. |
|---|---|---|---|---|
| Pair 1 | Country-before and country-after | 119 | 0.977 | 0.000 |

*5.2. Paired Samples t-Test*

In this context, we employed the paired samples *t*-test to clearly measure the difference in the mean that happened for the item groups. The results confirm that the emergence of the pandemic and precautionary measures in Saudi Arabia had significant effects on imported items, with *p*-value < 0.01 (see Table 7). We can say that there is a significant negative effect, as there is a huge difference among the means for each period, as shown in Table 8. The mean of the number of shipments imported for item groups before the emergence of COVID-19 is 11,764.42, while after the emergence of COVID-19, the mean becomes 6276.46, a decrease of 5487.96. Furthermore, the change rate decreases by 47%, which means that the number of shipments imported for each food item declines on average by close to half.

**Table 7.** Paired samples *t*-test result for item groups.

|  |  | t | df | Sig. (2-Tailed) |
|---|---|---|---|---|
| Pair 1 | Item_Before–Item_After | 4.030 | 23 | 0.001 |

**Table 8.** Paired samples statistics for item groups.

|  |  | Mean | N | Std. Deviation | Std. Error Mean |
|---|---|---|---|---|---|
| Pair 1 | Item_Before | 11,764.42 | 24 | 13,710.615 | 2798.667 |
|  | Item_After | 6276.46 | 24 | 7195.172 | 1468.708 |

We found that importing behavior clearly changed, as demonstrated in Figure 4.

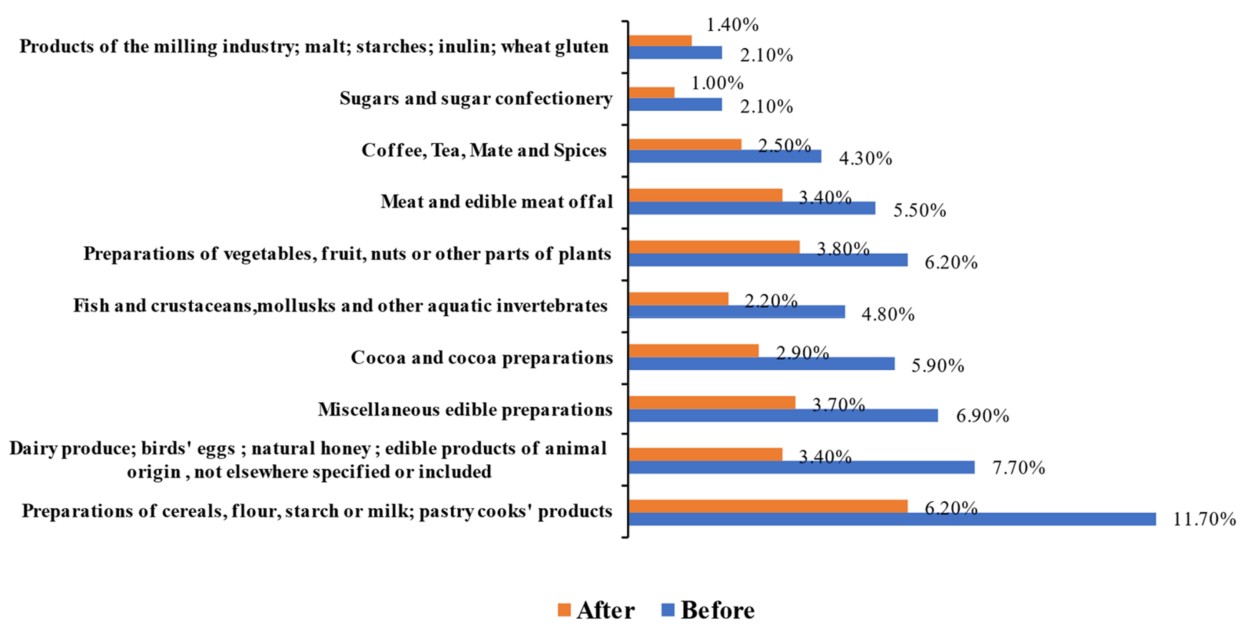

**Figure 4.** Top ten items: changes before and after the pandemic.

Furthermore, we used the same test to measure the change in behavior of the number of shipments imported from countries. The results show that the emergence of the pandemic

and precautionary measures in Saudi Arabia had a significant difference in the mean of the number of shipments imported from the studied countries, with *p*-value < 0.01 (Table 9).

**Table 9.** Paired samples *t*-test result for countries.

|  |  | *t* | **df** | **Sig. (2-Tailed)** |
|---|---|---|---|---|
| Pair 1 | Country-before–Country-after | 5.110 | 118 | 0.000 |

Consequently, there is a significantly negative effect because there is a clear difference among the means for each period, as shown in Table 10. The mean of the number of shipments imported from each country before the emergence of COVID-19 is 2372.66, while after the emergence of COVID-19, the mean becomes 1265.84, a decrease of 1106.82. Thus, the change rate declines by 47%. The behavior of countries from which the shipment was imported clearly changes, as shown in Figure 5.

**Table 10.** Paired samples *t*-test result for countries.

|  |  | **Mean** | **N** | **Std. Deviation** | **Std. Error Mean** |
|---|---|---|---|---|---|
| Pair 1 | Country-before | 2372.66 | 119 | 4787.214 | 438.843 |
|  | Country-after | 1265.84 | 119 | 2544.344 | 233.240 |

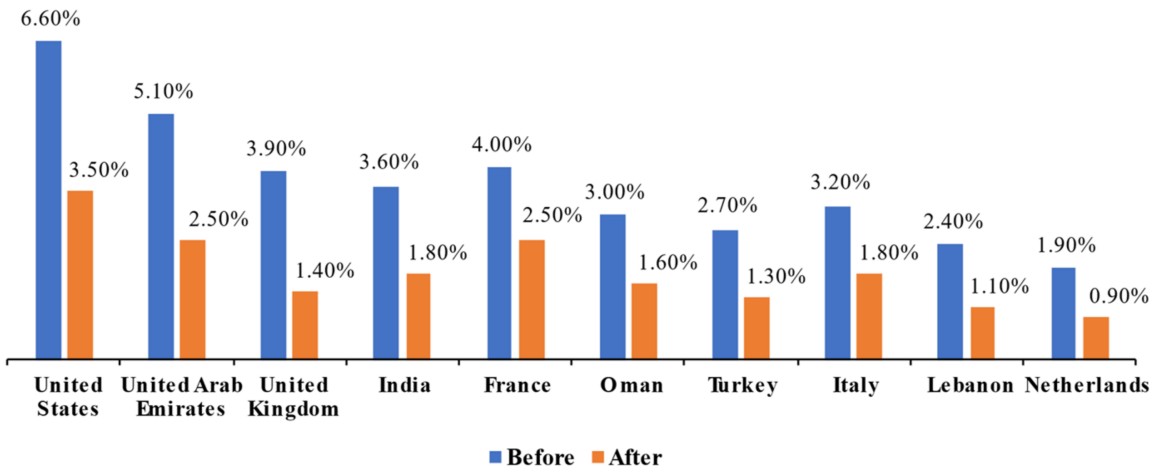

**Figure 5.** Top ten countries: number of shipments, changes before and after the pandemic.

### 5.3. Independent Samples t-Test for Item Groups

An additional test was performed to compare the mean of two independent populations to determine whether there is statistical evidence that the associated population means are significantly different for importing behavior for each item group before and after the emergence of COVID-19 in Saudi Arabia, using a 0.05 level of significance. By conducting the test, it was found that of the 25 item groups, 24 had a significantly different mean between the two periods (before and after emergence), and only one item group with significance > 0.05, which means the importing behavior of this item did not change (equal mean) between periods.

We found that the behavior clearly changed, as demonstrated in Table 11. The Levene's test in Table 11 was used to check the homogeneity of variance and prove the assumption of equal variance among groups [33], while the *t*-test for Equality of Means was applied to check the equality of the means of the two samples [34].

**Table 11.** Independent samples *t*-test for item groups.

| Item Group | Levene's Test for Equality of Variances | | *t*-Test for Equality of Means | | | |
|---|---|---|---|---|---|---|
| | F | Sig. | t | Df | Sig. (2-Tailed) | Mean Difference |
| Live animals | 5.162 | 0.023 | 1.979 | 757 | 0.048 | 0.028 |
| Meat and edible meat offal. | 50.863 | 0 | −10.106 | 610.265 | 0 | −26.832 |
| Fish and crustaceans, mollusks and other aquatic invertebrates. | 78.638 | 0 | −4.703 | 638.511 | 0 | −10.369 |
| Dairy produce; birds' eggs; natural honey; edible products of animal origin, not elsewhere specified or included. | 36.147 | 0 | −3.761 | 616.091 | 0 | −15.315 |
| Products of animal origin, not elsewhere specified or included. | 10.854 | 0.001 | 2.854 | 757 | 0.004 | 0.015 |
| Live trees and other plants; bulbs, roots and the like; cut flowers and ornamental foliage. | 0.024 | 0.877 | 0.167 | 1004 | 0.867 | 0.007 |
| Edible vegetables and certain roots and tubers. | 8.421 | 0.004 | −4.047 | 485.039 | 0 | −2.651 |
| Edible fruit and nuts; peel of citrus fruit or melons. | 21.297 | 0 | −4.790 | 609.43 | 0 | −2.750 |
| Coffee, Tea, Mate and Spices | 42.469 | 0 | −8.669 | 596.15 | 0 | −18.724 |
| Cereals | 49.496 | 0 | 7.534 | 622.024 | 0 | 7.413 |
| Products of the milling industry; malt; starches; inulin; wheat gluten. | 32.683 | 0 | 5.791 | 589.434 | 0 | 5.721 |
| Oil seeds and oleaginous fruits; miscellaneous grains, seeds and fruit; industrial or medicinal plants; straw and fodder | 24.329 | 0 | 8.163 | 534.438 | 0 | 3.037 |
| Lac; gums, resins and other vegetable saps and extracts. | 17.564 | 0 | 3.968 | 505.827 | 0 | 0.462 |
| Vegetable plaiting materials: vegetable products not elsewhere specified or included | 22.449 | 0 | 3.111 | 710.278 | 0.002 | 0.104 |
| Animal or vegetable fats and oil and their cleavage products; prepared edible fats; animal or vegetable waxes | 53.057 | 0 | 11.601 | 671.114 | 0 | 7.968 |
| Preparation of meat, of fish or of crustaceans, molluscs or other aquatic invertebrates | 95.996 | 0 | 15.864 | 871.802 | 0 | 11.154 |
| Sugars and sugar confectionery. | 85.41 | 0 | 15.116 | 793.473 | 0 | 20.268 |
| Cocoa and cocoa preparations | 58.366 | 0 | 11.881 | 708.561 | 0 | 40.504 |
| Preparations of cereals, flour, starch or milk; pastry cooks' products. | 70.948 | 0 | 14.844 | 656.096 | 0 | 76.043 |
| Preparations of vegetables, fruit, nuts or other parts of plants. | 266.229 | 0 | 22.015 | 1000.149 | 0 | 100.731 |
| Miscellaneous edible preparations | 90.867 | 0 | 14.654 | 723.981 | 0 | 47.461 |
| Beverages, spirits and vinegar | 47.292 | 0 | 9.941 | 635.592 | 0 | 6.183 |
| Residues and waste from the food industries; prepared animal fodder. | 5.387 | 0.02 | 2.004 | 645.958 | 0.046 | 0.165 |
| Salt; Sulphur, earths and stones; plastering materials, lime and cement | 7.197 | 0.007 | 2.828 | 518.84 | 0.005 | 0.385 |
| Albuminoidal substances; modified starches; glues; enzymes. | 13.968 | 0 | −4.854 | 361.937 | 0 | −0.445 |

*5.4. Comparison of Means Test for Item Groups*

After conducting an independent samples *t*-test, which showed that there is a significant difference in the means of 24 items, the Comparison of Means test was conducted to

determine the direction of this difference, whether it is increasing or decreasing. It was found that 17 item groups (68%) significantly decreased in their means after the emergence of COVID-19, while only 7 item groups (24%) significantly increased their means after the emergence of COVID-19, while one item (4%) remained the same. This was the same item that was referred to in the independent samples *t*-test, and its result was not significant (see Table 11), which confirms the validity of the results of the hypothesis test. Figure 6 shows the difference between the mean of the two periods for the top ten item groups with the largest differences in their means.

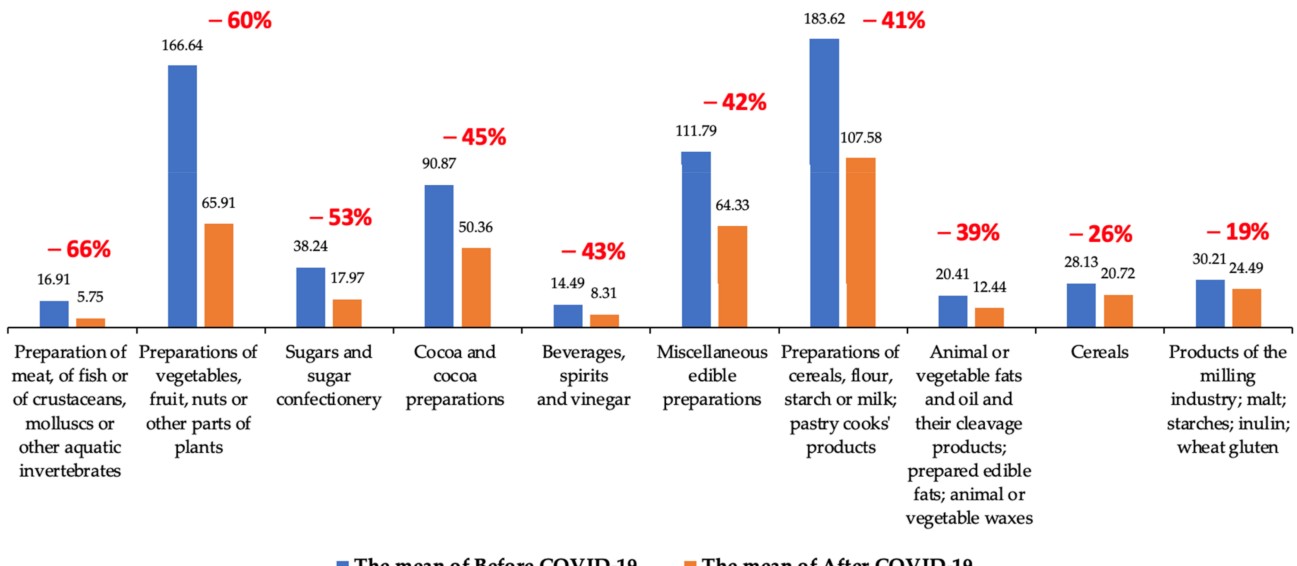

**Figure 6.** The top ten differences in mean for item groups.

### 5.5. Independent Samples t-Test for Countries

From the total of 127 countries in dataset, there are 86 countries (68%) for which the item groups imported from them show significantly different means before and after the emergence of COVID-19; only 41 countries (32%) show means which are not significantly different between the two periods.

### 5.6. Comparison of Means Test for Countries

The comparison of the means of countries shows that there are 83 countries (around 65%) with considerably reduced means after the emergence of COVID-19. Moreover, only 20 countries (16%) have means which significantly increased after the emergence of COVID-19, and only 24 (19%) remain the same in both periods. Figure 7 shows the differences between the means for the two periods of the top ten countries with the largest differences in their means.

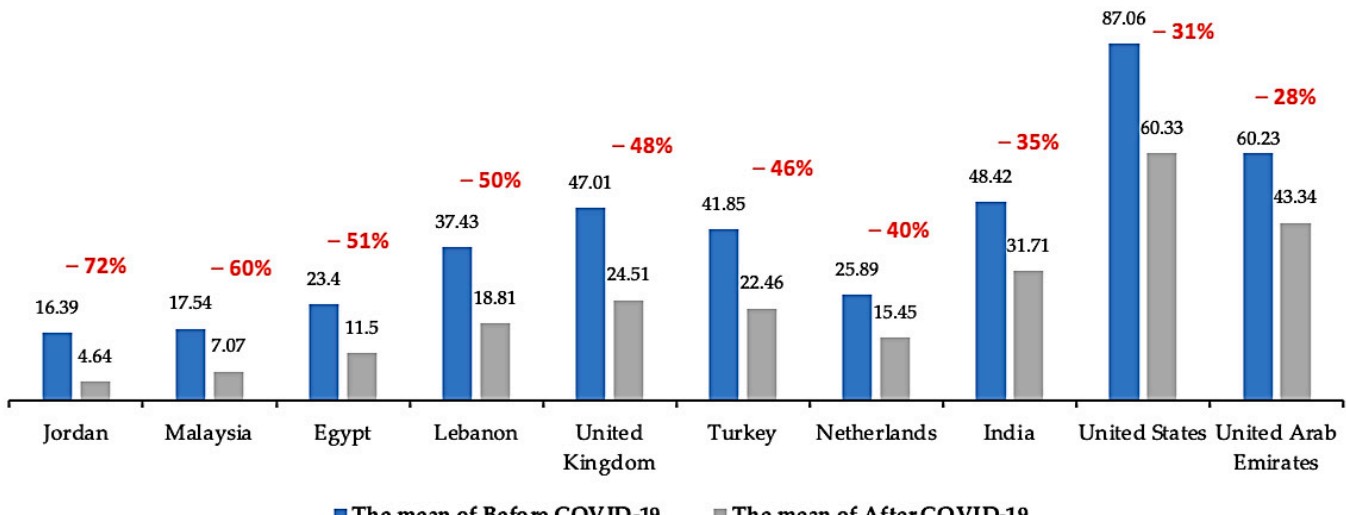

**Figure 7.** The top ten differences in mean for countries.

## 6. Conclusions and Recommendations

The present study aimed to investigate Saudi Arabia's food supply chain before and after the COVID-19 pandemic. The analysis shows that the burden of COVID-19 on food supply chains has clearly affected demand and caused a shortage of transportation. Evidently, the pandemic has negative impacts on food supply chains. The rate of change in importing activities over the study periods decreased by 47% for both item groups and countries. The correlation shows that the general distribution of item groups has not been affected by the emergence of the pandemic, and that the general distribution of countries from which imports are made has also not been affected. The paired sample *t*-test confirms that the emergence of the pandemic and precautionary measures in Saudi Arabia are having significant negative effects on imported items as well as on the countries from which items are imported. Moreover, the independent *t*-test shows that of the item groups and countries with significant differences between two periods, there are 84% of countries which show a significant difference in both periods, while 96% of item groups show a significant difference in their means. On the other hand, there are 86 countries where the imported item groups showed a notable change in their means, and 41 countries show minor changes between the two periods. Moreover, to find out the direction of this difference, we applied the comparison of means test and found that 17 item groups show a significant decrease in their means, while seven items (24%) show an increase and only one item (4%) retains the same value. Furthermore, the comparison shows that there are 83 countries with a significant decrease in their means, 20 countries where the averages increase significantly, and 24 which remain the same in both periods.

Saudi Arabia has already applied several strategies to achieve sustainability and overcome the impacts of the pandemic on its food supply chain without disrupting it, in consideration of the food security of its citizens. The country has overcome this crisis because it has worked on the goal of food security as one of the sustainability development goals in Vision 2030. To achieve sustainability in its food supply chain, Saudi Arabia aims to work on creating an optimal food supply source by conducting foreign investment in agricultural via direct partnerships with other countries on the strategic level [2]. Saudi Arabia increased the land available for farming by SR 1.9 billion (USD 506.67 million) in 2019 in order to build a sustainable food system by optimizing the use of agriculture lands [35], thereby helping the Agri-sector. Establishing a mitigation strategy to handle the impacts of the COVID-19 pandemic and linking the Ministry of Commerce, SFDA, Zakat, Tax and Customs Authority with the Ministry of Environment, Water, and Agriculture to work on developing this strategy in order to overcome upcoming issues is highly necessary, and making good decisions and taking the right actions in this regard are essential. One

recommendation is to use artificial intelligence and machine learning in order to help provide insights and forecast demand proactively. An additional technology that can play a major role in improving sustainability is using blockchain technology to enhance the traceability of food supply chains. In addition, performing what-if analysis to understand the current situation and investing in technology to improve food longevity by helping to maintain the shelf life of food is important. No doubt this research is also dependent upon two critical factors, transportation capacity and measuring the sustainable performance of the food supply chain. However, the evaluation of these two factors will be considered in our future work.

**Author Contributions:** Conceptualization, A.A.A., E.S. and A.K.J.S.; methodology, A.A.A. and E.S.; software, E.S.; validation, A.K.J.S., A.A. and M.A.; formal analysis, A.A.A.; investigation, M.B.K. and M.H.A.H.; resources, A.K.J.S. and A.A.A.; data curation, A.A.A.; writing—original draft preparation, A.A.A. and A.A.; writing—review and editing, A.K.J.S. and E.S.; visualization, M.B.K. and M.H.A.H.; supervision, A.K.J.S.; project administration, A.K.J.S.; funding acquisition, A.K.J.S. All authors have read and agreed to the published version of the manuscript.

**Funding:** This work was supported by the Deputyship for Research and Innovation, Ministry of Education in Saudi Arabia for funding this research work through the project number 959.

**Institutional Review Board Statement:** Not applicable.

**Informed Consent Statement:** Not applicable.

**Data Availability Statement:** Not applicable.

**Acknowledgments:** The authors would like to extend their appreciation to the Deputyship for Research and Innovation, Ministry of Education in Saudi Arabia for funding this research work through project number 959.

**Conflicts of Interest:** The authors declare that they have no conflict of interest to report regarding the present study.

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
