# Peer review of "Impacts of COVID-19 on the Food Supply Chain: A Case Study on Saudi Arabia"

_sustainability, doi:10.3390/su14010254_

Round 1
Reviewer 1 Report
The paper investigates the changes in food supply chains caused by the COVID-19 pandemic applying a solid statistical analysis of relevant data. The topic is current and interesting and definitely fits the journal's scope.
Even though the text is well-written, the proceeded data are relevant and the methodology is clearly presented, I suggest some modifications before publication.
As the authors themselves state, transportation is a key issue in supply chain operations and the pre-condition of freight transport access might also have a serious impact on food trade and operations.
However, the statistical analysis merely focuses on products and their trade before and during the pandemic. This means also a significant contribution but I suggest the investigation of transport capacity change in the COVID-19 period in further research (collecting data and their analysis would be impossible in the frames of a revision).
In the current paper, a more thorough discussion on the role of transportation is recommended based on some relevant references like:
-Ho et al (2021): The impact of COVID-19 on freight transport: Evidence from China. MethodsX, 8, 101200
-Loske (2020): The impact of COVID-19 on transport volume and freight capacity dynamics: An empirical analysis in German food retail logistics. Transportation Research Interdisciplinary Perspectives, 6, 100165
-Moslem et al (2020): Best-worst method for modelling mobility choice after COVID-19: evidence from Italy. Sustainability, 12(17), 6824
Apart from these citations, more references would improve the quality of the paper from relevant high-level journals.
Having amended the text and the list of references, I will recommend the paper for publication.
Author Response
To,
The Editor-in-Chief
Sustainability
Manuscript ID: sustainability-1509241
Note: The comments and suggestions made by the reviewers are gratefully acknowledged, kindly find below the point wise replies/corresponding amendments are made in the revised manuscript.
Reviewer 1 comments have been addressed in the revised version and response is attached.

Reviewer 2 Report
I do like the main idea of the paper and authors made significant work in literature review in order to create the list of indicators and analyse them.
However, in my opinion, the document is overloaded with information and needs more logical and focused structure especially in the first two chapters. The authors should separate the literature section.
Then for me it was not clear, why authors decided to concentrate on Saudi Arbia - I think some logical linking paragraphs must be added and mentioning of this region clearly in the paper.
There is a significant pull of papers that had been used for the development of framework, but it is not clear, how this list of papers have been obtained. Which keywords, databases have been used, inclusion/exclusion cretarie. It is important as based on it authors shows criteria. The authors may consider these recent studies to improve the quality of paper. https://doi.org/10.3390/en13246612 https://doi.org/10.1108/JABS-07-2021-0316 https://doi.org/10.3390/su12208747Author Response
To,
The Editor-in-Chief
Sustainability
Manuscript ID: sustainability-1509241
Note: The comments and suggestions made by the reviewers are gratefully acknowledged, kindly find below the point wise replies/corresponding amendments are made in the revised manuscript.
Reviewer 2 comments have been addressed in the revised version and response is attached.

Reviewer 3 Report
This case study aims to analyze the imported food during COVID-19 to understand how the food supply chain in Saudi Arabia is affected by this pandemic. Here has some issues should be addressed: 1. How the influences by pandemic were dealt with food supply chain sustainable development? 2. The references style in text is not consistent with the journal style. 3. The results have shown the food supply chain before and after the COVID-19 pandemic. However, there is no any about sustainable performance for food supply chain, the authors should add this evaluation results. 4. So what is the final results this paper have obtained? Only the statistical results? The conclusion is not clear presented in this paper. It seems more like a survey report. 5. How the impact obtained in this paper improves the further sustainability development in food supply chain and promotes the food shelf-life or quality assurance.Author Response
To,
The Editor-in-Chief
Sustainability
Manuscript ID: sustainability-1509241
Note: The comments and suggestions made by the reviewers are gratefully acknowledged, kindly find below the point wise replies/corresponding amendments are made in the revised manuscript.
Reviewer 3 comments have been addressed in the revised version and response is attached.

Round 2
Reviewer 1 Report
The authors have modified the paper accordingly, thus I recommend the publication.
Author Response
We would like to thank the reviewer for careful and thorough reading of this manuscript and for the thoughtful comments and constructive suggestions, which help to improve the quality of this manuscript.Reviewer 2 Report
Dear Authors,
Thank you for submitting the revised version of manuscript. I congratulate the authors for this interesting piece of work. Great job
Author Response
We would like to thank the reviewer for careful and thorough reading of this manuscript and for the thoughtful comments and constructive suggestions, which help to improve the quality of this manuscript.Reviewer 3 Report
The paper was revised well. There is a mistake in the manuscript section 1.1 is repeated. Please revised it.
Author Response
Thank you for your comment.
The paper has been revised and the section numbers have been changed.